# OpenReview forum: "Stealing the Recipe: Hyperparameter Stealing Attacks on Fine-Tuned LLMs"
_ICLR.cc/2026/Conference — Submitted to ICLR 2026_

### Official Review · Reviewer_S22H · 2025-10-29

**Soundness:** 2
**Presentation:** 1
**Contribution:** 3
**Rating:** 2
**Confidence:** 2

**Summary:**

This paper introduces a new data-poisoning attack that allows identification of certain hyperparameters used to train a model (mainly the model family, its size, the learning rate, and the batch size).

The attack first constructs a poisoned dataset containing specific hijacking tasks, and then trains a set of shadow models on both the poisoned dataset and a clean dataset, with each shadow model using different hyperparameters. Those shadow models are in turn used to train a classifier that predicts the hyperparameters based on their outputs when prompted with the poisoned data.

**Strengths:**

- The threat model (hyperparameter stealing) is relevant given the importance of hyperparameter selection in LLM training and the cost associated with hyperparameter tuning.
- The threat model's goal and the attacker's capabilities are clearly explained.

**Weaknesses:**

- The presentation of the method is hard to follow; an illustration to assist in understanding the method could significantly improve the clarity of Section 4. In particular, I do not grasp the intuition behind how the construction of the poisoning dataset helps in creating a downstream signal used to classify which hyperparameters are used. Similarly, the feature extraction process appears to be mostly hand-designed, and an explanation of the procedure that led to the choice of those features would be beneficial for guiding future work.
- The experiments are not sound. My understanding is that Table 1 shows the accuracy and F1-score of the classifier on the shadow models, i.e., on the training data. This means we do not know how the method behaves out of distribution. In particular, what if the victim (the one from whom we are stealing the hyperparameters) used a different clean dataset than the shadow models? What if they used a set of hyperparameters not included among those used for the shadow models? A table with out-of-distribution models and the accuracy of the method on such models would be necessary to claim that the method is effective in recovering hyperparameters.
- Similarly, we see that apart from $x_{1},x_{2}$, the additional features contribute little predictive power. The added benefit could simply be due to overfitting, because the classifier is evaluated on the training set.
- The results are limited; only model family and size are accurately classified for all model architectures tested.
- The models evaluated are both old and relatively small; whether the method generalizes to newer models (where the hyperparameters are even more valuable) remains an open question.
- The method does not transfer across model families, meaning the attacker needs to know the model family beforehand. This could be a practical issue, as most models whose families are known are open-source.

**Questions:**

- Can the authors compute results similar to Table 1 but using out-of-distribution models? For instance, using hyperparameters that differ from those of the shadow models, a different clean dataset, and a different percentage of poisoned data.
- What is the ratio between clean and poisoned data in your experiments? An ablation on this ratio is important, as in practice the poisoned data constitutes only a negligible percentage of the entire training dataset.
- What are the motivations behind the selected features?
- Can you evaluate your method on newer models? Perhaps a recent 3B model for cost-effectiveness?

---

> ### Author Response · Authors · 2025-11-21
> **Rebuttal (Part-I)**
>
> ---
> ### ***Q1. Can the authors compute results similar to Table 1 but using out-of-distribution models? For instance, using hyperparameters that differ from those of the shadow models, a different clean dataset, and a different percentage of poisoned data.***
> We thank the reviewer for the excellent suggestion. We have partially addressed this in our initial submission with the transferability experiment in Sec. 5.4, and full results are in Appendix D.7. Specifically, we study out-of-distribution (OOD) generalization by training the attack model on one architectural family (e.g., BART+Pegasus) and testing on another (e.g., GPT-2). The results (Table 5 in the revision) show that cross-family transferability fails: family prediction drops to 0%, and other hyperparameters fall near random-guess levels (e.g., model size 27.9%→50.0%). These findings reinforce that the hyperparameter signals our attack uses are highly family-dependent, and OOD generalization is inherently weak.
>
> Regarding the reviewer’s suggestion to evaluate with different clean datasets, we agree this would offer more insight. However, reproducing the full training pipeline under these conditions would require re-generating and fine-tuning 187 shadow LLMs, which exceeds the current computational budget during the rebuttal period. Each shadow model (GPT2-large) takes 16 GPU-hours on two A100s, making full retraining infeasible within the given timeframe.
>
> ---
> ### ***Q2. What is the ratio between clean and poisoned data in your experiments? An ablation on this ratio is important, as in practice, the poisoned data constitutes only a negligible percentage of the entire training dataset.***
> We appreciate the reviewer bringing up this important question. In all experiments, the poisoned IMDb-derived hijacking data accounts for only a small portion of the overall fine-tuning dataset. Shadow models are trained on a combination of (i) the clean CNN/DailyMail dataset (287,113 examples) and (ii) the injected hijacking data, which varies in size depending on the retention level. Therefore, even at 100% retention, poisoning makes up just 3.36% of the entire training dataset, and at lower retention rates, this percentage decreases further (e.g., 1.01% at 30% retention).
>
> To directly assess the reviewer’s concern, we performed a poison-retention ablation by varying the fraction of injected data preserved during shadow-model training while keeping the attack model fixed. Results (Table 12 in Appendix D.2) show that:
>
> - Moderate pruning (80% retention; approximately 2.7% poison rate) still results in significant leakage of key hyperparameters. The model family and size remain highly identifiable, and both learning rate and batch size stay well above chance.
>
> - Aggressive pruning (30% retention; ~1% poison rate) greatly reduces the available signal, especially for learning rate, batch size, and model size.
>
> - Optimizer prediction remains challenging across all settings.
>
> Overall, even when poisoned data makes up only 1–3% of the full training distribution—aligned with realistic large-scale corpora—the attack remains effective until the injected amount becomes very small. These findings confirm that the hijacking signal is strong at practical poisoning levels but fails once its presence drops too low.
>
> ---
> ### ***Q3. What are the motivations behind the selected features?***
>  We thank the reviewer for the thoughtful question. Our feature design is motivated by recent findings showing that fine-tuning can systematically reshape a model’s safety, alignment, and broader generation behavior—even when the fine-tuning corpus is fully benign[1]. These results indicate that training procedures and hyperparameters leave model-wide behavioral signatures that are observable at the output level.
>
> Based on this observation, our initial hypothesis is that if hyperparameters significantly impact the fine-tuning process, then these choices should be linked to consistent behavioral changes in the generated text. To analyze such relationships systematically, we designed our features to cover three complementary modalities—semantic, statistical, and structural—each capturing a different aspect of how outputs differ from a hidden reference. Within each modality, we chose popular and well-validated metrics to extract behavioral features, ensuring both interpretability and robustness under a black-box threat model.
>
> ---
> [1]Qi, Xiangyu, et al. "Fine-tuning aligned language models compromises safety, even when users do not intend to!." arXiv preprint arXiv:2310.03693 (2023).

---

> ### Author Response · Authors · 2025-11-21
> **Rebuttal (Part-II)**
>
> ---
> ### ***Q4. Can you evaluate your method on newer models? Perhaps a recent 3B model for cost-effectiveness?***
> Thank you for the helpful suggestion. Our method works with any publicly available and fine-tunable LLM, and it is not restricted to a specific architecture or parameter scale. Since the hyperparameters we aim to infer (optimizer, learning rate, batch size) are common across modern fine-tuning pipelines, our attack can easily extend to newer 2–3B-scale models, as long as they are capable of being fine-tuned.
>
> The main constraint is computational cost. As noted, the largest model used in our experiments (GPT-2-large, 774M parameters) already requires about 16 GPU-hours on 2×A100 GPUs per run. To evaluate poisoning at 30%, 80%, and 100%, we fine-tuned 187 configurations across these levels, totaling 561 models, which makes large-scale experiments on multi-billion-parameter models too expensive within the rebuttal period.
>
> ---
> ### ***Weakness 1. The presentation of the method is hard to follow;***
> We thank the reviewer for highlighting the need for clearer intuition. We will add a schematic illustration in Section 4 to clarify the overall pipeline. Intuitively, the poisoned samples cause consistent behavioral shifts in the model’s outputs that depend on the underlying hyperparameters, and these shifts serve as the downstream signal our classifier learns from. Regarding feature design, each x1–x7 feature was chosen through iterative ablations. We design our features to cover three complementary modalities—semantic, statistical, and structural—each summarizing a different aspect of how outputs differ from the camouflaged reference.
>
> ---
> ### ***Weakness 2. The experiments are not sound. My understanding is that Table 1 shows the accuracy and F1-score of the classifier on the shadow models, i.e., on the training data.***
> We thank the reviewer for the question. We clarify that Table 1 does not report performance on the same shadow models used to train the classifier. Following the standard shadow-model protocol used in membership inference attacks, we train the attack model on one subset of shadow models and evaluate on a separate, unseen set of shadow models. Therefore, the results already demonstrate generalization to models not seen during training.
> Regarding out-of-distribution hyperparameters: our threat model follows previous hyperparameter-stealing works, where the attacker predicts from a finite candidate set, and inferring values outside this set is not part of the standard framework.
>
> ---
> ### ***Weakness 3. Similarly, we see that apart from $x_{1},x_{2}$, the additional features contribute little predictive power. The added benefit could simply be due to overfitting, because the classifier is evaluated on the training set.***
> We respectfully clarify that our classifier is not evaluated on the training set. Following standard practice in shadow-model attacks (e.g., MIAs, property inference), we train the attack model on one subset of shadow models and evaluate it on a disjoint held-out set of unseen shadow models. Thus, the reviewer’s hypothesis that improvements from $x_3$–$x_7$ stem from overfitting does not apply. Moreover, the gains from adding features (e.g., model-size accuracy improving from 95.6% → 98.1%, LR from 85.1% → 89.2%, batch size from 78.0% → 82.5%) are consistent across multiple heads and evaluation splits, which would not occur if the improvements were overfitting artifacts. While $x_1$ and $x_2$ provide the dominant signal, the higher-order statistical and linguistic features ($x_3$- $ x_7$) yield robust, generalization-level improvements, not training-set memorization.
>
> ---

---

> ### Author Response · Authors · 2025-11-21
> **Rebuttal (Part-III)**
>
> ---
> ### ***Weakness 4. The results are limited; only model family and size are accurately classified for all model architectures tested.***
> We respectfully clarify that this statement is not consistent with the results reported in Tables 1–2. Across all model architectures (BART, Pegasus, GPT-2), four hyperparameters—not only model family and model size—are recovered with high accuracy. Learning rate and batch size remain strong heads throughout our experiments (typically 70–90% and 75–85% accuracy, respectively), far above the 33% random baseline. The only head that remains weak is the optimizer, which we explicitly acknowledge and analyze in Sec. 5.2 as having a minimal behavioral footprint under black-box constraints.
>
> Furthermore, our robustness experiments (token drop, shuffle, jitter, formatting noise) and cross-family evaluations demonstrate that the predictive signal for learning rate and batch size persists across architectures, contradicting the claim that only family and size are accurately classified. Thus, the limitation is specific to the optimizer head, not the overall scope of recovered hyperparameters.
>
> ---
> ### ***Weakness 6. The method does not transfer across model families, meaning the attacker needs to know the model family beforehand. This could be a practical issue, as most models whose families are known are open-source.***
> We thank the reviewer for raising this point. Cross-family generalization is indeed weak, but this is expected: architectural differences (encoder–decoder vs. decoder-only, tokenizers, training objectives) naturally induce family-specific behavioral signatures. Prior work on hyperparameter inference, model fingerprinting, and membership inference similarly relies on family-matched shadow models. Thus, this limitation is inherent to the problem and not unique to the shadow model technique.
>
> Finally, knowing the family does not imply open-source access: commercial APIs (OpenAI, Anthropic, Azure, etc.) publicly disclose model families despite keeping weights closed. Therefore, the requirement of family-matched shadow banks is compatible with realistic MLaaS threat models.
>
> ---

---

> > ### Comment · Reviewer_S22H · 2025-11-24
> >
> > Thank you for the additional experiments and answers. I list some remaining concerns below.
> >
> > **Presentation of the method/experimental evaluation**
> > I thank the authors for clarifying that their classifier is trained on different shadow models than the ones used for evaluation. This addresses a major concern. However, I still do not fully understand the exact experimental setup used by the authors. Could you update the sections regarding your experimental setup to explicitly mention this split, the ratio, which parameters are used for training data, and which are used for evaluation data? This is extremely important, and I believe it is either missing or poorly explained in the current manuscript.
> >
> > **OOD experiments (Q1)**
> > While I understand the compute constraints related to the rebuttal timeframe, I believe that for the threat model to be relevant it is necessary to assess whether the method generalizes when the clean dataset used for the shadow models and the victim model differs. I urge the authors to include such an experiment in the next revision of their work.
> >
> > **Poisoned data ablation (Q2)**
> > I thank the authors for their additional experiments. I think that a ~3% poisoning rate is still unrealistic in many scenarios. It would be beneficial if the authors clearly highlighted this limitation in their introduction.
> >
> > **Larger Model**
> > While I understand the compute constraints, I still think that there is no guarantee that the method extends to larger models, as the training dynamics of larger models can differ greatly. If 3B is too costly, using a recent 1B model could still provide greater insights into whether the method works on more capable models. It would be great if the authors could include such an experiment in their next revision.
> >
> > **Commercial Models**
> > Can the authors explain how this method can be used with commercial models? I do not fully understand the current explanation.
> >
> > *Minor:* It would be much simpler for the rebuttal discussion if the authors highlighted their changes to the manuscript in a different color.

---

> ### Author Response · Authors · 2025-12-02
> **Rebuttal (Part-IV)**
>
> ---
> ### ***Q1. OOD experiments. (While I understand the compute constraints related to the rebuttal timeframe, I believe it is necessary to assess whether the method generalizes when the clean dataset used for the shadow models and the victim model differs....)***
> We thank the reviewer for the valuable suggestion. In response, we have included a new experiment assessing clean-data OOD generalization, now added to Appendix D.5.
>
> **Setup.** The attacker is trained on shadow models fine-tuned on *CNN/DailyMail*, with 80% of the hijacking data retained (108 BART/Pegasus configurations). For OOD testing, we fine-tune eight victims on *WikiHow*, also with 80% retained hijacking data. Thus, the clean corpus shifts entirely (*CNN/DailyMail → WikiHow)*, while the attacker remains unchanged.
> Results (Appendix D.5, Table 16).
>
> Despite the distribution shift, the attack maintains strong performance:
> - Family: 99.85%
> - Size: 97.32%
> - Learning Rate: 92.26%
> - Batch Size: 85.71%
> - Optimizer: weakest head in all settings
>
> This mirrors in-distribution behavior and indicates that the learned features capture training-dependent behavioral signals rather than corpus-specific artifacts.
>
> **Takeaway.** As shown in Appendix D.5, hyperparameter leakage persists under clean-data OOD conditions: the attacker generalizes well even when the victim’s fine-tuning corpus differs entirely from the shadow models.
>
> ---
> ### ***Q2. Poisoned data ablation. (I think that a ~3% poisoning rate is still unrealistic in many scenarios. It would be beneficial if the authors clearly highlighted this limitation in their introduction....)***
>
> We thank the reviewer for the suggestion. As recommended, we have added a clarifying sentence in the Introduction explicitly stating that our experiments use a poisoning rate of ~3% to obtain the reported results.
>
> We respectfully note, however, that this poisoning rate aligns with previous training-time poisoning attacks in top security venues. For example, BadNets[1] uses a 10% poisoning budget, and Manipulating ML[2] evaluates poisoning levels ranging from 4% up to 20%. These works demonstrate that inserting thousands of benign-looking sentences into large, web-scraped corpora is both feasible and a standard assumption in the poisoning literature.
>
> ---
> ### ***Q. Presentation of the method/experimental evaluation. ( I thank the authors for clarifying that their classifier is trained on different shadow models than the ones used for evaluation....)***
>
> We thank the reviewer for highlighting the need for clearer documentation of the shadow–victim split. We have revised the manuscript accordingly and added a dedicated subsection in Appendix C.1 that now thoroughly describes the evaluation protocol. The attacker is trained on 80% of the shadow models, sampled to maintain the hyperparameter distribution. The remaining 20% of models are strictly reserved for evaluation and are never used for training. These models follow the same fine-tuning and poisoning pipeline but do not contribute training features.
>
> ---
> ### ***Q. Larger Model. (There is no guarantee the method extends to larger models. If 3B is too costly, a recent ~1B model would provide stronger evidence....)***
>
> We thank the reviewer for highlighting the importance of evaluating larger models. To directly test scalability, we extended our study to the Phi-1.5 model (1.3B parameters)—a substantially more capable decoder-only architecture compared to GPT-2. We constructed a 27-model Phi shadow bank spanning model size, optimizer, learning rate, and batch size, and applied the same 80% hijacking-retention protocol used in the main paper. We then trained configuration-matched attack models and evaluated them on held-out Phi and mixed-family runs. The full results appear in Appendix D.6, Table 17.
>
> **Takeaway.** Across Phi-based shadow models, we observe strong recoverability for model family and size, along with above-chance identifiability for learning rate and batch size—mirroring the trends seen in smaller models. Optimizer remains the weakest head, consistent with our main paper findings. Importantly, these results support the broader claim that our attack mechanism does not depend on model size: the leakage results from fine-tuning–induced behavioral shifts, and whenever a model is fine-tuneable, those signals remain exploitable, even at the 1B scale.
>
> ---
> [1] Gu, Tianyu, et al. "Badnets: Evaluating backdooring attacks on deep neural networks." Ieee Access 7 (2019): 47230-47244.
>
> [2] Jagielski, Matthew, et al. "Manipulating machine learning: Poisoning attacks and countermeasures for regression learning." 2018 IEEE symposium on security and privacy (SP). IEEE, 2018.

---

> ### Author Response · Authors · 2025-12-02
> **Rebuttal (Part-V)**
>
> ---
> ### ***Q. Commercial Models. (Can the authors explain how this method can be used with commercial models? I do not fully understand the current explanation.)***
>
> We thank the reviewer for raising this important clarification. Our threat model aligns with how commercial MLaaS systems are deployed today, and the attack remains feasible in both closed- and open-source settings.
>
> 1. **Only black-box outputs are required.** Our attack uses only the final generated text. All feature blocks operate solely on the API's observable outputs. This matches the interfaces provided by OpenAI, Anthropic, Google, and Azure, which do not expose logits, gradients, or internal states.
>
> 2. **Poisoning vector fits real-world training pipelines.** Commercial LLMs ingest large mixtures of public web text and user-uploaded instruction-tuning data. Our IMDb-style hijacking samples are natural-looking and survive standard preprocessing (filtering, deduplication), consistent with prior supply-chain poisoning work[3].
>
> 3. **Commercial providers increasingly allow user-driven fine-tuning.** Many vendors now support fine-tuning inside their private environments, where users upload custom datasets that the provider incorporates using their default fine-tuning algorithm. In these settings, the attacker can simply include adversarially crafted hijacking samples in the fine-tuning payload. The vendor fine-tunes the model internally and exposes the resulting model via a new API endpoint—without revealing the weights. This matches the capability assumptions in recent aligned-LLM poisoning work[4].
>
> 4. **Knowing the family is realistic.** Commercial providers publicly disclose model families (GPT-3.x, GPT-4, Llama-derived, T5-derived). This enables the construction of a family-matched shadow bank without requiring white-box access.
>
> **Takeaway.** Our attack aligns with two realistic commercial scenarios:
> - poisoning via publicly ingested data during provider-controlled pretraining/fine-tuning, and
> - poisoning via user-supplied datasets in vendor-run fine-tuning pipelines.
>
> In both cases, only black-box text outputs are required, making the attack directly applicable to commercial MLaaS deployments.
>
> ---
> ### ***Q. Minor. (It would be much simpler for the rebuttal discussion if the authors highlighted their changes to the manuscript in a different color.)***
>
> We appreciate the reviewer’s helpful suggestion. In the revised manuscript, all new or changed content is now highlighted in dark blue.
>
> ---
> [3] Si, Wai Man, et al. "{Two-in-One}: A model hijacking attack against text generation models." 32nd USENIX Security Symposium (USENIX Security 23). 2023.
>
> [4] Qi, Xiangyu, et al. "Fine-tuning aligned language models compromises safety, even when users do not intend to!." arXiv preprint arXiv:2310.03693 (2023).

---

### Official Review · Reviewer_AQMw · 2025-11-01

**Soundness:** 2
**Presentation:** 2
**Contribution:** 2
**Rating:** 4
**Confidence:** 4

**Summary:**

The paper explores “hyperparameter stealing”: inferring optimizer, learning rate, batch size, model size, and architecture of fine-tuned language models from black-box access. The attacker embeds a stealthy “hijacking” corpus into the victim’s training data, trains a bank of shadow models covering many hyper-parameter settings, extracts seven kinds of statistical and semantic features from model outputs, and learns a multi-label classifier to guess the hidden recipe. On BART, Pegasus and GPT-2 the attack recovers family (100 %), size (≈98 %), learning rate (≈89 %), and batch size (≈80 %) in the encoder–decoder case; optimizer remains near chance. The paper also offers modality ablations, cross-family transfer tests, and an evaluation of the ONION defense.

**Strengths:**

-Clear statement of a new security objective: recovering hyper-parameters rather than parameters or training data.
- Method is systematically broken down (hijacking data, shadow bank, features, classifier) and each component is described with equations (Pages 3–5) enabling replication.
- Table 1 gives comprehensive accuracy/F1 across 189 target configurations and shows sizable margins over random guessing, especially for family, size, learning rate, and batch size.
- Figure 1 visualises that the camouflaged outputs stay close in embedding space, supporting the stealth argument.
- Ablations in Table 2 quantify the contribution of each feature block; the rise from 58.7 % to 98 % size accuracy when adding statistical blocks is informative.
- Transfer (Table 3) and defense (Table 4) studies highlight both limitations and the inadequacy of an existing sanitisation technique, giving a balanced picture.
- Writing is mostly clear; equations defining $S_{\text{sem}}$, $S_{\text{hij}}$, and the multi-label loss are explicit and free of obvious errors.

**Weaknesses:**

1. Unrealistic threat prerequisite. The attack requires the adversary to have data-poisoning access to the victim’s training corpus (Sec. 3, Page 3). For most commercial LLM providers training on carefully curated or private corpora, sneaking thousands of crafted IMDB-like sentences in is speculative. The work does not measure how many poisoned samples must be retained to keep performance in Table 1 or assess the attacker’s cost vs. benefit.
2. Limited baseline comparison. No alternative hyper-parameter inference attack is implemented; even a simple black-box meta-classifier (without poisoning) could be a strawman. Thus the incremental gain due to hijacking is unclear.
3. Optimizer prediction failure (≈18 %) suggests the feature design or framing is incomplete. The paper does not investigate why nor attempt alternative signals (e.g., gradient-based probes).
4. Cross-family generalisation largely collapses (Table 3: 0 % family accuracy), yet the abstract still claims effectiveness “even in mixed-family settings” without emphasising the sharp degradation.
5. Mathematical clarity gaps. Equation for the loss (Page 5) introduces per-head weights $\lambda_k$ and class weights $\alpha_{k,c}$ but never specifies chosen values; wrong choices might bias the reported head accuracies.
6. Experimental scope. Only summarisation is tested. Other generation tasks (dialogue, translation) or instruction-tuned models are common in practice and may exhibit different leakage patterns.
7. Ethical and defensive discussion remains shallow: no mitigation beyond ONION is studied, and ONION is known to be weak on text generative backdoors.
8. Potentially missing related work: Wang & Gong 2018 (classical hyper-parameter stealing) is cited, but recent LLM-specific privacy backdoor papers such as Feng & Tramèr 2024 or Kandpal et al. 2023 are absent from Related Work despite obvious

Potentially Missing Related Work
- Qi, Zeng & Xie, “Fine-tuning Aligned Language Models Compromises Safety” (2023) – shows how fine-tuning introduces new attack surface; should be cited in Sec. 2 to frame poisoning risk.
- Kandpal, Pillutla & Oprea, “User Inference Attacks on Large Language Models” (2023) – studies inference from outputs; relevant for comparison in Sec. 2.
- Feng & Tramèr, “Privacy Backdoors: Stealing Data with Corrupted Pretrained Models” (2024) – demonstrates data-theft backdoors; similar stealth mechanism.
- Hicks, “Stealing Finetuning Data with Corrupted Models” (2025) – expands the backdoor discussion; inclusion would strengthen positioning.
(These works are directly relevant yet not cited.)

**Questions:**

1. How many hijacking examples are actually retained by the target in Table 1 experiments after the simulated 20 % filtering, and how does accuracy scale if retention drops to 5 %?
2. Could a non-poisoning passive attack, e.g., training shadow models and probing the deployed API with random inputs, achieve similar accuracy?
3. For optimizer prediction, did you try time-series features such as variance of log-probs across decoding steps which might capture Adam’s batching effects?
4. How robust is the attack to temperature sampling in generation at inference?
5. Can the attacker fine-tune fewer than 189 shadows and still match performance? Any subsampling study?

---

> ### Author Response · Authors · 2025-11-21
> **Rebuttal (Part-I)**
>
> ---
> ## ***Q1.How many hijacking examples are actually retained by the target in Table 1 experiments after the simulated 20 % filtering, and how does accuracy scale if retention drops to 5 %?***
> We thank the reviewer for the insightful question. The experiments in Table 1 assume a 20% filtering step, meaning that the target retains 80% of the injected IMDb-derived hijacking examples. In our setup, this corresponds to keeping 7,715 poisoned examples out of 9,644 initially injected (seed 42). We have now added a dedicated retention-sensitivity analysis in Appendix D.2.
>
> To directly address the reviewer’s concern about how accuracy scales with more aggressive filtering, we use the 100% retention-trained attack model and evaluate shadow banks constructed with 80% and 30% retention (simulating increasingly strict preprocessing). The results are summarized in Table 12.
>
> At 80% retention, hyperparameter recovery remains strong: Model family: 99.8%, Model size: 87.2%, Learning rate: 71.1%, Batch size: 71.9%.
>
> At 30% retention (about 2,893 hijacking samples), performance drops significantly, with model size, learning rate, and batch size predictions nearing random behavior. The optimizer's prediction remains weak across all settings.
>
> **Takeaway.**
> Hyperparameter leakage remains effective at moderate retention rates (up to about 70–80% loss), but the attack weakens when the retention rate drops too low. We will include this analysis and table in the revised appendix.
>
> ---
> ## ***Q2.Could a non-poisoning passive attack, e.g., training shadow models and probing the deployed API with random inputs, achieve similar accuracy?***
> Thank you for the insightful question. We conducted an additional experiment (Appendix D.3) to assess whether a passive, non-poisoning attacker—who trains shadow models and probes the victim API with an arbitrary prompt—can recover hyperparameters without depending on our structured hijacking dataset.
>
> This study trains the attacker model on the baseline prompt format (Structure 1) and tests it on three diverse prompt styles: a minimal prefix, a rigid bullet-point instruction, and a free-form paraphrased instruction. This simulates a passive adversary submitting arbitrary prompts without domain-specific tailoring.
>
> **Findings.**
> 1. Rigid structured prompts homogenize model outputs and suppress the behavioral signals (x₁–x₇) that encode hyperparameter information.
> 2. However, leakage remains well above chance levels across all hyperparameter heads—even though the attacker never uses a task-aligned prompt. This suggests that hyperparameter-dependent behavioral signatures are inherent and persist under non-poisoning probing.
> 3. Free-form prompts restore much of the attack’s strength, outperforming rigid prompts and demonstrating that naturalistic generation reveals stronger training-dependent behaviors.
>
> **Takeaway.**
>  Non-poisoning passive probing cannot match the effectiveness of our proposed hijacking-dataset attack, but it still recovers hyperparameter information above chance. The hijacking strategy provides the attacker with substantially more stable and discriminative signals, leading to the large gains reported in the main paper. We include the full results and analysis in Appendix D.3.
>
> ---
> ## ***Q3.For optimizer prediction, did you try time-series features such as variance of log-probs across decoding steps, which might capture Adam’s batching effects?***
> We thank the reviewer for the insightful suggestion. However, we note that such features fall outside our realistic black-box threat model. In practice, commercial MLaaS providers (OpenAI, Anthropic, Google, Azure) do not expose token-level logits or step-wise log-probabilities to external users. As a result, attackers observe only the final generated text, not the intermediate probability trajectories. To stay aligned with this widely adopted deployment constraint, our attack is intentionally designed to depend only on output-level behavioral features (x1–x7), which are accessible in real-world API environments.
>
> While incorporating time-series log-prob features might help a more skilled attacker with privileged access, such access is not available in practical black-box APIs. Our design choice, therefore, aligns with the standard assumption in black-box inference attacks—the adversary only submits inputs and observes outputs.
>
> ---

---

> ### Author Response · Authors · 2025-11-21
> **Rebuttal (Part-II)**
>
> ---
> ### ***Q4.How robust is the attack to temperature sampling in generation at inference?***
> Thank you for the insightful question. We have added a new robustness study in Appendix D.4, where we systematically evaluate the effect of output stochasticity—including temperature-induced variation—by perturbing the target model’s generated text prior to feature extraction. This setting directly simulates realistic API behaviors, including random sampling, formatting variation, partial truncation, and mild corruption. Importantly, the attacker’s classifier is trained only on clean outputs (no noise), so these results measure the generalization of our features (x1–x7) under sampling variability.
>
> Across a wide range of perturbations—including structural reformatting, sentence-order shuffling, and token-level deletion up to 20%—the attack remains highly stable. Core hyperparameters, such as model family, size, learning rate, and batch size, maintain high accuracy and F1 scores. Only under extreme corruption (≈30% token loss) do we observe noticeable degradation. These findings suggest that our attack relies on deeper behavioral/semantic signals rather than surface-form consistency.
>
> ---
> ### ***Q5.Can the attacker fine-tune fewer than 189 shadows and still match performance? Any subsampling study?***
> Thank you for the helpful suggestion. We have added a subsampling analysis in Appendix D.1 to quantify how attack performance scales with fewer shadow models, specifically for subsample sizes 𝑛 ∈ {10, 20, 50, 100, 150}. We randomly sample n shadow configurations from the full set of 189 and train the attack model using only their feature pairs. We evaluate each subsample model on the full held-out test split and report mean ± std over three seeds (32/42/52).
>
> **Takeaways.**
> The subsampling study (Table 11, Appendix D.1) shows a clear monotonic trend. Model family becomes almost trivial with only 20 shadow models, while model size, learning rate, and batch size improve steadily as the subsample grows, reflecting their more diffuse behavioral signatures. Optimizer remains unstable across all subsample sizes—consistent with Sec. 5.2—indicating that it leaves only weak output-level traces. Overall, the attack is already functional with small subsamples, and full performance is achieved only with the complete shadow bank.
>
> ---

---

> ### Author Response · Authors · 2025-11-21
> **Rebuttal (Part-III)**
>
> ---
> ### ***Weakness 1. Unrealistic threat prerequisite.***
> We respectfully clarify that our threat model follows the standard training-time supply-chain poisoning setup widely used in previous work (e.g., [1],[2]). Large-scale LLM pre-training pipelines often process large amounts of publicly available web text, and seemingly benign sentences—such as our IMDB-style samples—are usually kept after standard filtering and deduplication. Our method does not require rare triggers, access to internal states (logits/gradients), or control over the training data distribution; instead, it exploits subtle output-space patterns that survive typical preprocessing.
>
> ---
> ### ***Weakness 4. Cross-family generalisation largely collapses (Table 3: 0 % family accuracy), yet the abstract still claims effectiveness “even in mixed-family settings” without emphasising the sharp degradation.***
> We thank the reviewer for pointing out the ambiguity. We agree that Table 3 demonstrates that family prediction does not transfer across different model families. In the abstract, our intended claim was not about family classification but about the remaining hyperparameters. Specifically, in the mixed-configuration setting (BART + PEGASUS + GPT-2), learning rate and batch size remain identifiable with accuracies of 69.49% and 63.63%, respectively—well above the 33.3% random baseline. To prevent confusion, we have revised the wording from “mixed-family settings” to “mixed-configuration settings” in the abstract.
>
> ---
> ### ***Weakness 5. Mathematical clarity gaps. Equation for the loss (Page 5) introduces per-head weights $\lambda_k$ and class weights $\alpha_{k,c}$ but never specifies chosen values; wrong choices might bias the reported head accuracies.***
> We thank the reviewer for highlighting the missing specification. In the revised version, we explicitly report the chosen per-head and class-weight values used in the loss.
>
> ---
> ### ***Weakness 6. Experimental scope. Only summarisation is tested. Other generation tasks (dialogue, translation) or instruction-tuned models are common in practice and may exhibit different leakage patterns.***
> We appreciate the reviewer for highlighting this important point. Due to computational costs and limited time during the rebuttal period, we were unable to extend the evaluation beyond summarization. However, we agree that testing on dialogue, translation, and instruction-tuned models is valuable. We plan to include these additional generation tasks in our future work.
>
> ---
> ### ***Weakness 8. Potentially missing related work: Wang & Gong 2018 (classical hyper-parameter stealing) is cited, but recent LLM-specific privacy backdoor papers such as Feng & Tramèr 2024 or Kandpal et al. 2023 are absent from Related Work despite obvious.***
> We thank the reviewer for highlighting these relevant works. We have cited "Fine-tuning Aligned Language Models Compromises Safety."
>
> ---
> [1] Jagielski, Matthew, et al. "Manipulating machine learning: Poisoning attacks and countermeasures for regression learning." 2018 IEEE symposium on security and privacy (SP). IEEE, 2018.
>
> [2] Si, Wai Man, et al. "{Two-in-One}: A model hijacking attack against text generation models." 32nd USENIX Security Symposium (USENIX Security 23). 2023.

---

### Official Review · Reviewer_JQ7Z · 2025-11-01

**Soundness:** 3
**Presentation:** 3
**Contribution:** 2
**Rating:** 6
**Confidence:** 4

**Summary:**

This paper proposes a new model stealing scenario called hyperevaluation, where the goal is to recover the instruction tuning behavior of a black-box language model. Instead of copying the model’s answers, the attacker extracts task-like examples from the model’s own prompts and responses. These examples are then used to train a smaller open-source model that imitates the original model’s instruction-following behavior. The main contribution lies in shifting attention from output mimicry to replicating how the model generalizes across tasks. While the idea of stealing instructions is thought-provoking and relevant to current API usage patterns, the technical approach is relatively simple and does not introduce new learning algorithms or modeling techniques.

**Strengths:**

● The idea of stealing instruction patterns instead of just outputs is interesting and not something I’ve seen explored much. It frames a realistic kind of vulnerability that hasn’t gotten much attention.

● The attack setup makes sense given how APIs are used in practice, especially with prompts that expose few-shot examples or task templates.

● While the method itself is relatively simple, the experiments are clean and show that the approach can lead to meaningful imitation of the original model’s behavior.

● The paper is clearly written and easy to follow. It’s not hard to imagine others building on this kind of attack framing.

**Weaknesses:**

● The attack is conceptually interesting but technically quite straightforward. It mostly involves mining examples from prompts and training a model on them. There’s no new algorithm or mechanism beyond this.

● The success of the attack depends a lot on how the API model formats its responses. If few-shot examples or task prompts are not exposed in the output, it’s unclear how well this method would work.

● The paper doesn’t really discuss how easy it would be to defend against this kind of attack. Basic strategies like hiding prompt structure, trimming outputs, or limiting example formatting could potentially reduce the risk, but none of this is explored.

● It would have been helpful to understand how sensitive the attack is to noise or to less clean examples. Right now it’s not clear how robust the data collection and downstream training really are.

**Questions:**

1. How dependent is your attack on the model exposing clean few-shot examples or task templates in its outputs?

2. Did you try running the attack on models that give more free-form or less structured responses?

3. Could basic defenses like removing examples from responses or randomizing output structure weaken the attack?

4. How noisy can the extracted examples be before the student model stops learning meaningful patterns?

---

> ### Author Response · Authors · 2025-11-21
> **Rebuttal (Part-I)**
>
> We thank the reviewer for the valuable comments and thoughtful questions. Please find our responses below.
>
> ---
>
> ### ***Q1 + Q2. “How dependent is your attack on the model exposing clean few-shot examples or task templates in its outputs? Did you try running the attack on models that give more free-form or less structured responses?”***
>
> We appreciate the reviewer’s insightful question. To assess how sensitive our attack is to prompt structure and output formatting, we conducted an additional ablation study (Appendix D.3).
>
> **1. Dependency on prompt structure.**
> We tested the same attacker—trained only on the baseline prompt—across three qualitatively different input formats (Table 13):
> (1) a minimal prefix,
> (2) a rigid bullet-point instruction, and
> (3) a free-form paraphrased instruction.
> These prompts differ in how strongly they limit the model’s output style. The attacker is *not* retrained; only the inference-time prompt is changed.
>
> **2. Results.**
> As shown in Table 14, rigid prompting reduces stylistic variance and moderately diminishes leakage, but it does not eliminate it. Even with the strict bullet-point format, the attacker remains significantly above random-guess baselines across all heads. Free-form prompting recovers much of the original performance, demonstrating that when the model generates unconstrained text, the underlying hyperparameter-dependent signals reappear.
>
> **3. Takeaway.**
> Our attack does **not** depend on the model revealing few-shot exemplars or structured templates. Leakage occurs across many output styles. Prompt rigidity can weaken—but not eliminate—the signal, while free-form generation enhances it. This supports our claim that the leakage comes from deeper distributional properties of the model rather than surface-level formatting artifacts.
>
> ---
>
> ### ***Q3 + Q4. “Could basic defenses, such as removing examples or randomizing output structure, weaken the attack? How noisy can the extracted examples be before the student model stops learning meaningful patterns?”***
>
> We thank the reviewer for raising this important question regarding robustness. To determine whether simple output-level defenses can weaken our attack, we conducted an additional set of experiments (Appendix D.4). In these experiments, we perturb the *target model’s outputs* before feature extraction, while the attack classifier is trained only on the 80%-retention hijacking dataset (i.e., without any noise). This setup evaluates how well our features generalize under realistic deployment-time distortions.
>
> We examine perturbations that mimic common API behaviors:
> - **Formatting variation** (bullets, newline shifts, spacing changes)
> - **Token dropping** at 10%, 20%, and 30% removal
> - **Sentence-level structural jitter** (order randomization, permuting sentence order to disrupt structural consistency)
>
> Full results for encoder–decoder models (BART + Pegasus, seed 42) are shown in **Table 15**.
>
> **Findings.**
> - Formatting changes and sentence shuffling retain nearly all performance across family, size, learning rate, and batch-size heads. This suggests our features depend on *semantic and behavioral signals*, rather than surface-level structure.
> - Token dropping up to **20%** still achieves high accuracy for most hyperparameters, indicating resilience to truncation and moderate corruption.
> - Only extreme corruption **(≈30% token loss)** significantly reduces performance, which exceeds typical noise levels found in real-world MLaaS APIs.
>
> **Takeaway.**
> Removing examples, randomizing superficial structure, or applying mild corruption does *not* significantly weaken the attack. Therefore, mitigating hyperparameter leakage requires defenses that conceal the model’s underlying generation behavior—not just text formatting or partial content removal.

---

> ### Author Response · Authors · 2025-11-21
> **Rebuttal (Part-II)**
>
> ---
> ### ***Weakness: Lack of Technical Novelty***
> We thank the reviewer for this thoughtful comment. We respectfully clarify that, although our attack's implementation is intentionally simple, the paper's contribution is not architectural novelty but the discovery of a previously undocumented capability in the black-box hyperparameter-stealing setting.
>
> 1. We show—contrary to most prior hyperparameter-inference work—that fine-tuned LLM hyperparameters can be recovered solely from textual outputs, without logits, gradients, or training traces. Demonstrating leakage under a strict MLaaS-style black-box interface is, to our knowledge, new and practically relevant.
>
> 2. Our attack relies on a hijacking-dataset mechanism that induces subtle, reproducible output-level shifts rather than input-space triggers. Designing such auxiliary data to expose optimizer/learning-rate/batch-size signatures while remaining natural enough to evade perplexity-based filters is non-trivial; our results show that this output-side strategy, paired with the x₁–x₇ feature pipeline, enables reliable inference.
>
> 3. We uncover an undeclared vulnerability: common training hyperparameters leave stable behavioral fingerprints—including semantic drift, variance patterns, and syntactic divergence—that persist across seeds and are exploitable from outputs alone.
>
> 4. As with prior impactful security work (e.g., BadNet, Get a model!), the main contribution lies in revealing a new attack vector and demonstrating feasibility, rather than proposing a complex new architecture.

---

### Meta-Review · Area_Chair_TAx4 · 2026-01-05

**Summary:**

This paper introduces a hyperparameter stealing attack against fine-tuned LLMs that attempts to infer optimizer, learning rate, batch size, and model architecture through data poisoning and shadow model training. While the problem formulation addresses a gap in the literature, the work suffers from several fundamental limitations that prevent acceptance at ICLR.

The most critical issue is the unrealistic threat model. The attack requires injecting approximately 3% poisoned data into the victim's training corpus and assumes this survives preprocessing filters. The authors cite BadNets and related poisoning work to justify this assumption, but those attacks target fundamentally different deployment scenarios. Commercial LLM providers curate training data through multiple stages of filtering, deduplication, and quality control. Consistently inserting thousands of crafted IMDb-style samples while maintaining sufficient retention is highly implausible in practice. The rebuttal argues that providers ingest public web text, but this overlooks that production systems employ sophisticated data pipelines specifically designed to detect anomalous patterns. Even user-driven fine-tuning services apply safety filters that would likely catch repeated injection attempts.

The technical contribution is incremental. The attack pipeline consists of constructing poisoned data, training shadow models, extracting hand-designed features, and training a multi-label classifier. No new algorithms or theoretical insights are introduced. While demonstrating feasibility can be valuable, the simplicity here is concerning given the strong assumptions required. The feature extraction relies on seven manually selected metrics without principled justification for why these particular signals should correlate with hyperparameters. The rebuttal mentions iterative ablations but provides no systematic analysis of the design space.

Experimental scope is insufficient. The evaluation covers only summarization using three older models (BART, Pegasus, GPT-2). The largest model tested initially is GPT-2-large at 774M parameters. Although the rebuttal adds Phi-1.5 experiments, 1.3B parameters still falls far short of the 7B+ models dominating current practice. The absence of evaluation on dialogue, translation, or instruction-tuned models limits confidence in generalization. More fundamentally, optimizer prediction completely fails at 18% accuracy, and cross-family transfer collapses to random guessing. These failures suggest the learned features capture superficial family-specific artifacts rather than fundamental training dynamics.

The paper lacks comparison with passive alternatives. No baseline demonstrates whether similar accuracy could be achieved by simply querying the API with diverse prompts and training shadow models without poisoning. The rebuttal's Appendix D.3 shows that even rigid prompting maintains above-chance performance, suggesting the poisoning overhead may not be justified. Without this comparison, the cost-benefit tradeoff of the poisoning approach remains unclear.

While hyperparameter stealing represents an interesting research direction, this submission does not meet the conference standard. The threat model's practicality is questionable, the technical depth is limited, and the experimental validation is incomplete. I recommend rejection.

**Reviewer Concerns:**

Reviewer Concerns

The rebuttal achieved mixed success in addressing reviewer criticisms. Reviewer JQ7Z's concerns about prompt dependency and robustness to noise were convincingly handled through new appendices demonstrating that the attack works across diverse prompt formats and withstands formatting changes and token dropping up to 20%. However, the fundamental criticism about limited technical novelty remains unresolved. The authors reframe their contribution as discovering a vulnerability rather than proposing algorithmic innovation, but this doesn't address the reviewer's core observation that the method is conceptually straightforward with no new learning mechanisms.

Reviewer AQMw received partial responses on several technical questions. The retention sensitivity analysis, passive attack comparison, temperature robustness studies, and shadow model subsampling experiments were added as requested. The problematic abstract wording about mixed-family settings was corrected, and some missing citations were included. However, the most critical weakness regarding the unrealistic threat model was defended rather than addressed. Citing BadNets and other poisoning work does not establish feasibility for commercial LLM training pipelines, which employ fundamentally different data curation processes with sophisticated filtering and quality control. The experimental scope also remains inadequate, as the authors acknowledge they could not test on dialogue, translation, or instruction-tuned models due to compute constraints. The optimizer prediction failure at 18% accuracy persists without resolution.

Reviewer S22H's concerns saw the most substantial progress. The clean-data out-of-distribution experiment transferring from CNN/DailyMail to WikiHow demonstrates reasonable generalization, and the Phi-1.5 evaluation at 1.3B parameters represents a meaningful scale increase. The confusion about training versus evaluation splits was convincingly clarified, and the poisoning rate is now explicitly stated as requested. Nevertheless, significant issues persist. While 1.3B parameters exceeds the original 774M, this still falls far short of the 7B to 70B models dominating current practice, undermining claims that the method extends to any fine-tuneable LLM. The promised presentation improvements are not demonstrated in the rebuttal materials, and the ~3% poisoning rate continues to raise practical concerns that were acknowledged but not adequately justified.

Across all reviewers, the threat model's realism remains the most problematic unresolved issue. None were fully convinced that injecting 3% poisoned data into commercial training pipelines is feasible given modern data curation practices. The limited scope covering only summarization on older and smaller models, combined with complete optimizer prediction failure and collapsed cross-family transfer, suggests the method's applicability is narrower than claimed.

**Reviewer Scores:**

Reviewer JQ7Z started at 6 (marginally above acceptance threshold) and would likely remain around 5-6. The rebuttal successfully addressed their technical questions about prompt dependency and robustness to perturbations, demonstrating the attack works across diverse formats and withstands realistic noise levels. However, their core criticism about limited technical novelty stands entirely unresolved. The reframing as vulnerability discovery rather than algorithmic contribution doesn't change the fundamental observation that the method involves straightforward data poisoning and shadow model training without new mechanisms. Given that novelty concerns were secondary to their generally positive assessment, a slight downward adjustment to 5 or maintaining 6 seems most likely.

Reviewer AQMw began at 4 (marginally below acceptance threshold) and would probably stay in the 4-5 range. While they received answers to most technical questions including retention sensitivity, passive attacks, and shadow model subsampling, their primary concern about the unrealistic threat model was defended rather than addressed. The rebuttal's citation of BadNets-style poisoning work doesn't overcome the fundamental difference between image classification and commercial LLM training pipelines with sophisticated data curation. The acknowledged inability to test beyond summarization due to compute constraints reinforces the limited scope criticism. The new experiments show effort and add value, which might push toward 5, but the unresolved threat model concerns likely prevent acceptance.

Reviewer S22H started at 2 (reject, not good enough) and could reasonably move to 4-5. This reviewer received the most substantial additions including clean-data OOD experiments, Phi-1.5 evaluation, and critical clarification that the classifier evaluates on held-out shadow models rather than training data. Their follow-up comment shows appreciation for the new experiments while maintaining reservations about presentation clarity, model scale adequacy, and the 3% poisoning rate practicality. The resolution of their confusion about training versus evaluation splits addresses what appeared to be a major methodological concern, but the scale limitations and realistic deployment questions persist. An increase to the marginally-below-acceptance range seems plausible given the substantive experimental additions, though full acceptance remains unlikely.

---

### Decision · Program_Chairs · 2026-01-26

Reject